# Contribution to the Synthesis, Characterization, Separation and Quantification of New *N*-Acyl Thiourea Derivatives with Antimicrobial and Antioxidant Potential

**DOI:** 10.3390/pharmaceutics15102501

**Published:** 2023-10-20

**Authors:** Roxana Roman, Lucia Pintilie, Diana Camelia Nuță, Miron Teodor Căproiu, Florea Dumitrașcu, Irina Zarafu, Petre Ioniță, Ioana Cristina Marinaș, Luminița Măruțescu, Eleonora Kapronczai, Simona Ardelean, Carmen Limban

**Affiliations:** 1Department of Pharmaceutical Chemistry, Faculty of Pharmacy, “Carol Davila” University of Medicine and Pharmacy, 6 Traian Vuia, 020956 Bucharest, Romania; roxana.roman@drd.umfcd.ro (R.R.); diana.nuta@umfcd.ro (D.C.N.); carmen.limban@umfcd.ro (C.L.); 2National Institute of Chemical-Pharmaceutical Research & Development, 112 Vitan Av., 031299 Bucharest, Romania; 3“C. D. Nenitzescu” Institute of Organic and Supramolecular Chemistry, 202B Splaiul Independenței, 060023 Bucharest, Romania; dorucaproiu@gmail.com (M.T.C.); fdumitra@yahoo.com (F.D.); 4Department of Organic Chemistry, Biochemistry and Catalysis, Faculty of Chemistry, University of Bucharest, 4-12 Regina Elisabeta, 030018 Bucharest, Romania; zarafuirina@yahoo.fr (I.Z.); p_ionita@yahoo.co.uk (P.I.); 5Research Institute of the University of Bucharest, University of Bucharest, 90 Panduri Road, 030018 Bucharest, Romania; ioana.cristina.marinas@gmail.com; 6Sanimed International Impex S.R.L., 087040 Calugareni, Romania; lumi.marutescu@gmail.com; 7Department of Microbiology and Immunology, Faculty of Biology, University of Bucharest, 91-96 Splaiul Independenței, 060101 Bucharest, Romania; 8Supramolecular Organic and Organometallic Chemistry Centre, Department of Chemistry, Faculty of Chemistry and Chemical Engineering, Babeş-Bolyai University, 11 Arany János, 400028 Cluj-Napoca, Romania; 9Department of Pharmaceutical Technology, Faculty of Pharmacy, “Vasile Goldiș” Western University, 86 Liviu Rebreanu, 310045 Arad, Romania; simonaaardelean@yahoo.com

**Keywords:** *N*-acyl thiourea derivatives, antimicrobial, antibiofilm, antioxidant activity, validation, HPLC, ICH

## Abstract

The present study aimed to synthesize, characterize, and validate a separation and quantification method of new *N*-acyl thiourea derivatives (**1a**–**1o**), incorporating thiazole or pyridine nucleus in the same molecule and showing antimicrobial potential previously predicted in silico. The compounds have been physiochemically characterized by their melting points, IR, NMR and MS spectra. Among the tested compounds, **1a**, **1g**, **1h**, and **1o** were the most active against planktonic *Staphylococcus aureus* and *Pseudomonas aeruginosa*, as revealed by the minimal inhibitory concentration values, while **1e** exhibited the best anti-biofilm activity against *Escherichia coli* (showing the lowest value of minimal inhibitory concentration of biofilm development). The total antioxidant activity (TAC) assessed by the DPPH method, evidenced the highest values for the compound **1i,** followed by **1a**. A routine quality control method for the separation of highly related compounds bearing a chlorine atom on the molecular backbone (**1g**, **1h**, **1i**, **1j**, **1m**, **1n**) has been developed and validated by reversed-phase high-performance liquid chromatography (RP—HPLC), the results being satisfactory for all validation parameters recommended by the ICH guidelines (i.e., system suitability, specificity, the limits of detection and quantification, linearity, precision, accuracy and robustness) and recommending it for routine separation of these highly similar compounds.

## 1. Introduction

Many heterocyclic compounds containing thiophene, pyrazole, thiazolidine, *s*-triazine, pyridazine, fluoroquinolone, benzothiazole, benzoxazolone nuclei as well as thiourea moiety have proven to imprint versatile biological activities. They have been reported to possess antimicrobial [1,2,3,4,5,6], anticancer [7,8,9], antiviral [10,11], anti-allergic [12], anticonvulsant [13,14,15,16,17], and antioxidant activities [2,18].

The introduction of atoms, functional groups or other heterocyclic cores to the thiourea pharmacophore structure has been proven to be a beneficial approach for improving some of the desired biological activities. Previous studies have demonstrated that binding of heterocyclic rings to the thiourea moiety lowers the toxicity and increases their potency. In this regard, promising anti-HIV agents were obtained by attaching **1h**-imidazole moiety to thiourea derivatives [19,20]. Also, in a series of new 1-benzoyl, 3-phenyl-thiourea derivatives, *halo*- and *methoxy*-groups substituted on aryl rings displayed increased activity against all tested bacterial strains, compared to the other substituents [21].

Some *N*-(4-*R*-phenyl)/benzyl/(2-phenylethyl)-*N*′-(6-phenylpyridazin-3-yl)thiourea derivatives carrying thiourea group in position 3 were synthesized and evaluated for their antimicrobial activity using a broth microdilution test. These compounds showed promising inhibitory activity against *Staphylococcus aureus* and *Escherichia coli* bacterial strains and antifungal activity against *Candida* sp. [4].

A few thiourea derivatives of (2′-(**1h**-tetrazol-5-yl)biphenyl-4-yl)methanamine demonstrated in vitro antibacterial activity against *Staphylococcus aureus*, *Pseudomonas aeruginosa*, and *Escherichia coli*. These compounds also exhibited important antifungal activity [6].

Novel 2-methylquinazolin-4(3*H*)-one derivatives bearing thiourea functionality in position 3 were synthesized and screened for their antimicrobial and anti-inflammatory (TNF-α and IL-6) activities. Some of the compounds showed moderate antibacterial activities in comparison with ciprofloxacin [22].

New compounds containing thiourea group in position 6 of 3-methyl-2(3*H*)-benzoxazolone and 5-chloro-3-methyl-2(3*H*)-benzoxazolone rings were evaluated and 1-phenyl-3-(5-chloro-3-methyl-2-oxo-3*H*-benzoxazole-6-yl)thiourea and 1-benzyl-3-(5-chloro-3-methyl-2-oxo-3*H*-benzoxazole-6-yl)thiourea exhibited good inhibitory profiles against *Escherichia coli* [23].

New thiourea derivatives of 1,3-thiazole have been synthesized and tested in vitro against Gram-positive cocci, Gram-negative strains, *Candida albicans*, *Mycobacterium tuberculosis* reference (H37Rv), and clinical strains. 1-(3,4-Dichlorophenyl)-3-(1,3-thiazol-2-yl)thiourea and 1-(3-chloro-4-fluorophenyl)-3-(1,3-thiazol-2-yl)thiourea showed a significant inhibitory effect against Gram-positive cocci with MIC values between 2 and 32 µg/mL. The same compounds inhibited the biofilm formation of both methicillin-susceptible and resistant *S. epidermidis* strains. For the antimicrobial activity, the presence of the halogen atom, especially in the third position of the phenyl nucleus is crucial. All these compounds have also proved to be cytotoxic on the MT-4 tumoral cells and exhibited antiviral activity on a large set of DNA and RNA viruses, including Human Immunodeficiency Virus (HIV) type 1 [10].

Inspired by the data collected from the scientific literature and based on the molecular descriptors and molecular docking studies carried out in our previous work [24], the present study aimed to synthesize new *N*-acyl thiourea derivatives (referred to in the current paper as **1a**–**1o**), that incorporate both thiazole or pyridine nuclei in the same molecule. The presence of electron-withdrawing, electron-donating atoms, and specific functional groups on the acyl thiourea moiety’s heterocyclic core has been investigated to elucidate their influence on antimicrobial, antibiofilm, and antioxidant activities.

Among the synthesized compounds, six molecules were designed by adding the chlorine atom to the molecular backbone, leading to close structural similarity of the resulting chemical entities. For this reason, a routine quality control method for the separation of the highly related compounds (**1g**, **1h**, **1i**, **1j**, **1m**, **1n**) has been developed and validated through reversed-phase high-performance liquid chromatography method (RP—HPLC).

## 2. Materials and Methods

### 2.1. Chemistry

General procedure for the synthesis of the new compounds (**1a**–**1o**):

A solution of ammonium thiocyanate (0.01 mol) in anhydrous acetone (5 mL) was added to a solution of 2-(4-ethylphenoxymethyl)benzoyl chloride (0.01 mol) in anhydrous acetone (15 mL). The reaction mixture was refluxed for one hour. A solution of the primary heterocyclic amine in anhydrous acetone was added to the cooled mixture. Then, the reaction mixture was heated for two hours. The resulting compound was precipitated by pouring it into cold water. The crude substance was purified by crystallization from 2-propanol using charcoal.

The synthesis of the 2-((4-ethylphenoxy)methyl)benzoic acid and 2-((4-ethylphenoxy)methyl)benzoic acid chloride was presented in a previous article [25].

### 2.2. Measurements

Most of the reagents were used as received from Sigma-Aldrich (St. Louis, MO, USA) and Merck (Darmstadt, Germany). Ammonium thiocyanate was dried at 100 °C, and acetone was dried on calcium chloride and then distilled.

#### 2.2.1. Melting Points

Melting points were recorded by Electrothermal 9100 apparatus (Bibby Scientific Ltd., Stone, UK) and were uncorrected.

#### 2.2.2. Infrared Spectra

The IR spectra were recorded on a Bruker Vertex 70 FT-IR spectrometer (Bruker Corporation, Billerica, MA, USA). Absorptions were reported with the following relative intensities: w—weak band; m—medium band; s—intense band; vs—very intense band.

#### 2.2.3. Nuclear Magnetic Resonance Spectra

**^1h^** NMR and ^13^C NMR spectra were recorded on a Bruker Fourier 300 MHz instrument (Bruker Corporation, Billerica, MA, USA), operating at 300 MHz for **^1h^** NMR and at 75 MHz for ^13^C NMR in deuterated dimethyl sulfoxide (DMSO-d6), with tetramethylsilane used as internal standard. Data were reported as follows: a chemical shift in ppm (δ), multiplicity (s = singlet, brs = broad singlet, d = doublet, t = triplet, q = quartet, m= multiplet, dd = double doublet, td = triple doublet, ddd = doublet of doublets), signal/atom attribution, and the coupling constant (Hz). For ^13^C NMR data, the order is as follows: chemical shifts and signal/atom attribution; Cq = quaternary carbon.

#### 2.2.4. Mass Spectra

The APCI + high-resolution mass spectra (MS) were recorded on a Thermo Scientific LTQ-Orbitrap XL spectrometer with a standard ESI/APCI source. Thermo Xcalibur 4.0 software was used to process the mass spectra (XcaliburTM Software, Thermo Fisher Scientific, 168 3rd Avenue, Waltham, MA USA, www.thermofisher.com).

#### 2.2.5. Spectral Data

2-((4-Ethylphenoxy)methyl)-*N*-((thiazol-2-yl)carbamothioyl)benzamide (**1a**)

(m.p. 125–126.5 °C; yield 72%, 2.86 g)

**^1h^**-NMR (DMSO-d6, δ ppm, *J* Hz, T = 303 K): 12.60(brs, 2H, NH); 7.69(dd, **1h**, H-4, 1.4, 7.5); 7.62(dd, **1h**, H-4, 1.4, 7.5); 7.56(td, **1h**, H-5, 7.5, 1.4); 7.52(d, **1h**, H-18, 3.7); 7.46(td, **1h**, H-6, 7.5, 1.4); 7.26(d, **1h**, H-19, 3.7); 7.05(d, 2H, H-11, H-13, 8.6); 6.81(d, 2H, H-10, H-14, 8.6); 5.28(s, 2H, H-8); 2.51(q, 2H, H-15, 7.5); 1.12(t, 3H, H-15′, 7.5) (Appendix A, for **^1h^**-NMR spectrum)

^13^C-NMR (DMSO-d6, δ ppm, T = 303 K): 178.05(C-16); 166.72(C-1); 158.17(C-17); 156.23(C-9); 137.53(C-18); 136.21(Cq); 136.05(Cq); 133.06(Cq); 130.78(C-5); 128.54(C-11, C-13); 128.53(C-4); 128.42(C-7); 127.67(C-6); 114.64(C-10, C-14); 113.65(C-19); 67.29(C-8); 27.20(C-15); 15.75(C-15′) (Appendix A, for ^13^C-NMR spectrum).

FT-IR (ATR in solid, ν cm^−1^): 3222w; 3111w; 3058w; 2961m; 2914m; 1667s; 1607m; 1576m; 1546vs; 1507vs; 1439m; 1380w; 1280s; 1235vs; 1183m; 1161m; 1129m; 1035s; 892w; 818m; 731m; 702m; 673m; 614w (Appendix A, for IR spectrum).

Chemical Formula: C_20_H_19_N_3_O_2_S_2_, Exact Mass: 397.09187, HRMS (APCI+, DMSO + MeOH): *m*/*z* calcd for C_19_H_19_N_2_O_2_S^+^ = 339.11618, found 339.11537 (36%, mass error ∆m = −2.42 ppm), calcd for C_16_H_5_O_2_^+^ = 239.10666, found 239.10603 (100%, ∆m = −2.63 ppm), calcd for C_1**1h**9_N_2_OS^+^ = 217.04301, found 217.04246 (78%, ∆m = −2.35 ppm).

2-((4-Ethylphenoxy)methyl)-*N*-((pyridin-2yl)carbamothioyl)benzamide (**1b**)

(m.p. 68.3–70.1 °C; yield 81%, 3.17 g)

**^1h^**-NMR (DMSO-d6, δ ppm, *J* Hz, T = 303 K): 13.06(s, **1h**, NH, deuterable); 11.99(brs, **1h**, NH, deuterable); 8.74(brs, **1h**, H-21); 8.40(brs, **1h**, H-18); 7.89(td, **1h**, H-19, 6.9, 2.0); 7.65(dd, **1h**, H-4, 1.2, 7.4); 7.60(dd, **1h**, H-7, 1.8, 7.4); 7.57(td, **1h**, H-6, 7.5, 1.2); 7.47(td, **1h**, H-6, 7.5, 1.8); 7.26(td, **1h**, H-20, 6.9, 2.0); 7.03(d, 2H, H-11, H-13, 8.6); 6.87(d, 2H, H-10, H-14, 8.6); 5.28(s, 2H, H-8); 2.47(q, 2H, H-15, 7.5); 1.08(t, 3H, H-15′, 7.5) (Appendix A, for **^1h^**-NMR spectrum).

^13^C-NMR (DMSO-d6, δ ppm, T = 303 K): 177.67(C-16); 170.08(C-1); 156.17(C-9); 151.15(Cq); 148.29(C-18); 137.96(C-19); 135.95(Cq); 133.21(Cq); 131.10(C-5); 128.64(C-4); 128.51(C-11, C-13); 128.39(C-7); 127.71(C-6); 121.22(C-20); 115.37(CH); 114.55(C-10, C-14); 67.44(C-8); 27.20(C-15); 15.69(C-15′) (Appendix A, for ^13^C-NMR spectrum).

FT-IR (ATR in solid, ν cm^−1^): 3457m; 3236w; 3030m; 2958m; 2921m; 2861w; 1674m; 1612m; 1576m; 1514vs; 1456s; 1433s; 1387m; 1327s; 1290s; 1240vs; 1155s; 1030m; 838w; 812m; 779m; 734m; 688m; 655w (Appendix A, for IR spectrum).

Chemical Formula: C_22_H_21N3_O_2_S, Exact Mass: 391.13545, HRMS (APCI+, DMSO + MeOH): *m*/*z* calcd for [M + H]^+^ C_22_H_22_N_3_O_2_S^+^ = 392.14272, found 392.14171 (100%, ∆m = −2.58 ppm) calcd for [M − H]^+^ C_22_H_20_N_3_O_2_S^+^ = 390.12707, found 390.12683 (84%, ∆m = −0.62 ppm), calcd for C_16_H_18_NO_2_^+^ = 256.13321, found 256.13301 (41%, ∆m = −0.78 ppm), calcd for C_16_H_15_O_2_^+^ = 239.10666, found 239.10655 (74%, ∆m = −0.46 ppm), calcd for C_8_H_8_NO^+^ = 134.06004, found 134.06000 (6%, ∆m = −0.30 ppm).

2-((4-Ethylphenoxy)methyl)-*N*-((pyridin-3-yl)carbamothioyl)benzamide (**1c**)

(m.p. 157.6–159 °C; yield 63%, 2.46 g)

**^1h^**-NMR (DMSO-d6, δ ppm, *J* Hz): 12.32(s, **1h**, NH, deuterable); 11.97(brs, **1h**, NH, deuterable); 8.63(d, **1h**, H-21, 2.3); 8.44(dd, **1h**, H-20, 1.6, 4.8); 8.01(ddd, **1h**, H-18, 1.6, 2.3, 8.3); 7.63(dd, **1h**, H-4, 1.2, 7.4); 7.60(dd, **1h**, H-7, 1.8, 7.4); 7.57(td, **1h**, H-6, 7.5, 1.2); 7.47(td, **1h**, H-6, 7.5, 1.8); 7.44(dd, **1h**, H-19, 4.6, 8.3); 7.09(d, 2H, H-11, H-13, 8.6); 6.90(d, 2H, H-10, H-14, 8.6); 5.28(s, 2H, H-8); 2.51(q, 2H, H-15, 7.5); 1.12(t, 3H, H-15′, 7.5) (Appendix A, for **^1h^**-NMR spectrum).

^13^C-NMR (DMSO-d6, δ ppm): 180.23(C-16); 169.99(C-1); 156.27(C-9); 147.00(C-20); 146.14(C-21); 136.20(Cq); 135.84(Cq); 134.87(Cq); 133.25(Cq); 132.41(C-18); 131.02(C-5); 128.57(C-11, C-13); 128.45(C-4); 128.34(C-7); 127.71(C-6); 123.25(C-18); 114.55(C-10, C-14); 67.47(C-8); 27.23(C-15); 15.76(C-15′) (Appendix A, for ^13^C-NMR spectrum).

FT-IR (ATR in solid, ν cm^−1^): 3128w; 3032w; 2956m; 2912m; 2868m; 1673m; 1530vs; 1510vs; 1382m; 1284m; 1233s; 1199m; 1168s; 1120m; 1029m; 881w; 821m; 746s; 707m; 675m; 648w; 603w (Appendix A, for IR spectrum).

Chemical Formula: C_22_H_21N3_O_2_S, Exact Mass: 391.13545, HRMS (APCI+, DMSO + MeOH): *m*/*z* calcd for [M + H]^+^ C_22_H_22_N_3_O_2_S^+^ = 392.14272, found 392.14227 (100%, ∆m = −1.15 ppm), calcd for [M − S − H]^+^ C_22_H_20_N_3_O_2_^+^ = 358.15500, found 358.15497 (4%, ∆m = −0.08 ppm), calcd for C_16_H_18_NO_2_^+^ = 256.13321, found 256.13303 (12%, ∆m = −0.70 ppm), calcd for C_16_H_15_O_2_^+^ = 239.10666, found 239.10638 (38%, ∆m = −1.17 ppm), calcd for C_6_H_5_N_2_S^+^ = 137.01680, found 137.01651 (6%, ∆m = −2.12 ppm), calcd for C_8_H_8_NO^+^ = 134.06004, found 134.05977 (8%, ∆m = −2.01 ppm).

2-((4-Ethylphenoxy)methyl)-*N*-((3-methylpyridin-2-yl)carbamothioyl)benzamide (**1d**)

(m.p. 112–113.3 °C; yield 71%, 2.88 g)

**^1h^**-NMR (DMSO-d6, δ ppm, *J* Hz): 12.11(s, **1h**, NH, deuterable); 11.94(brs, **1h**, NH, deuterable); 8.31(dd, **1h**, H-21, 4.5, 1.3); 7.73(dd, **1h**, H-19, 1.3, 7.6); 7.63(dd, **1h**, H-4, 1.2, 7.4); 7.60(dd, **1h**, H-7, 1.8, 7.4); 7.57(td, **1h**, H-5, 7.5, 1.2); 7.47(td, **1h**, H-6, 7.5, 1.8); 7.30(dd, **1h**, H-20, 4.5, 7.6); 7.09(d, 2H, H-11, H-13, 8.6); 6.89(d, 2H, H-10, H-14, 8.6); 5.27(s, 2H, H-8); 3.31(s, 3H, H-18′); 2.51(q, 2H, H-15, 7.5); 1.14(t, 3H, H-15′, 7.5) (Appendix A, for **^1h^**-NMR spectrum).

^13^C-NMR (DMSO-d6, δ ppm): 180.02(C-16); 169.87(C-1); 156.21(C-9); 150.04(C-17); 146.15(C-C-21); 133.39(C-19); 136.18(Cq); 135.78(Cq); 133.34(Cq); 130.99(C-5); 130.53(Cq); 128.64(C-4); 128.59(C-11, C-13); 128.53(C-7); 127.74(C-6); 123.20(C-20); 114.66(C-10, C-14); 67.48(C-8); 27.24(C-15); 17.03(C-18′); 15.77(C-15′) (Appendix A, for ^13^C-NMR spectrum).

FT-IR (ATR in solid, ν cm^−1^): 3149m; 3062w; 2960m; 2929m; 2870m; 1672s; 1604w; 1581w; 1504vs; 1454vs; 1373m; 1326m; 1235s; 1161s; 1114m; 1019m; 954w; 882w; 816w; 734m; 667m; 582w (Appendix A, for IR spectrum).

Chemical Formula: C_23_H_23_N_3_O_2_S, Exact Mass: 405.15110, HRMS (APCI+, DMSO + MeOH): *m*/*z* calcd for [M + H]^+^ C_23_H_24_N_3_O_2_S^+^ = 406.15837, found 406.15779 (31%, ∆m = −1.43 ppm), calcd for [M − H]^+^ C_23_H_22_N_3_O_2_S^+^ = 404.14272, found 404.14235 (47%, ∆m = −0.92 ppm), calcd for C_16_H_18_NO_2_^+^ = 256.13321, found 256.13296 (21%, ∆m = −0.98 ppm), calcd for C_16_H_15_O_2_^+^ = 239.10666, found 239.10636 (100%, ∆m = −1.25 ppm), calcd for C_7_H_7_N_2_S^+^ = 151.03245, found 151.03220 (28%, ∆m = −1.66 ppm), calcd for C_8_H_8_NO^+^ = 134.06004, found 134.05982 (22%, ∆m = −1.64 ppm).

2-((4-Ethylphenoxy)methyl)-*N*-((4-methylpyridin-2-yl)carbamothioyl)benzamide (**1e**)

(m.p. 92.6–93.9 °C; yield 68%, 2.76 g)

**^1h^**-NMR (DMSO-d6, δ ppm, *J* Hz): 13.01(s, **1h**, NH, deuterable); 11.94(brs, **1h**, NH, deuterable); 8.60(brs, **1h**, H-18); 8.24(brs, **1h**, H-21); 7.63(dd, **1h**, H-4, 1.2, 7.4); 7.60(dd, **1h**, H-7, 1.8, 7.4); 7.57(td, **1h**, H-5, 7.5, 1.2); 7.47(td, **1h**, H-6, 7.5, 1.8); 7.10(brs, **1h**, H-20); 7.09(d, 2H, H-11, H-13, 8.6); 6.89(d, 2H, H-10, H-14, 8.6); 5.27(s, 2H, H-8); 2.49(q, 2H, H-15, 7.5); 2.36(s, 3H, H-19′); 1.09(t, 3H, H-15′, 7.5) (Appendix A, for **^1h^**-NMR spectrum).

The formation of hydrogen bonds inhibits the free rotation around the SC–N–C^17^ bond and instead, the protons in the pyridine ring are on the time scale of the NMR experiment. By heating to 70 °C, the respective signals become sharper and present a hyperfine structure. The signals of carbon atoms in the pyridine ring appear broadened for the same reason.

^13^C-NMR (DMSO-d6, δ ppm): 177.52(C-16); 170.14(brs, C-1); 156.16(C-9); 151.28(C-17); 148.93(C-19); 147.90(C-21); 136.11(Cq); 135.92(Cq); 133.26(Cq); 131.08(C-5); 128.51(C-7); 128.50(C-11, C-13); 128.39(C-4); 127.71(C-6); 122.18(C-20); 115.79(C-18); 114.54(C-10, C-14); 67.42(C-8); 27.18(C-15); 20.88(C-19′); 15.67(C-15′) (Appendix A, for ^13^C-NMR spectrum).

FT-IR (ATR in solid, ν cm^−1^): 3348w; 3162w; 3035w; 2968w; 2922w; 2872w; 1675m; 1606m; 1507vs; 1446m; 1378m; 1326s; 1289m; 1225s; 1151s; 1026m; 897w; 828m; 793w; 735m; 668m; 596w (Appendix A, for IR spectrum).

Chemical Formula: C_23_H_23_N_3_O_2_S, Exact Mass: 405.15110, HRMS (APCI+, DMSO + MeOH): *m*/*z* calcd for [M + H]^+^ C_23_H_24_N_3_O_2_S^+^ = 406.15837, found 406.15710 (100%, ∆m = −3.13 ppm), calcd for [M − H]^+^ C_23_H_22_N_3_O_2_S^+^ = 404.14272, found 404.14241 (8%, ∆m = −0.78 ppm), calcd for C_16_H_18_NO_2_^+^ = 256.13321, found 256.13269 (2%, ∆m = −2.03 ppm), calcd for C_16_H_15_O_2_^+^ = 239.10666, found 239.10630 (6%, ∆m = −1.51 ppm), calcd for C_7_H_7_N_2_S^+^ = 151.03245, found 151.03225 (2%, ∆m = −1.32 ppm).

2-((4-Ethylphenoxy)methyl)-*N*-((5-methylpyridin-2-yl)carbamothioyl)benzamide (**1f**)

(m.p. 128–129.3 °C; yield 85%, 3.44 g)

**^1h^**-NMR (DMSO-d6, δ ppm, *J* Hz, T = 333 K): 12.86(s, **1h**, NH, deuterable); 11.82(brs, **1h**, NH, deuterable); 8.54(brs, **1h**, H-18); 8.23(brs, **1h**, H-21); 7.77(dd, **1h**, H-19, 1.9, 8.6); 7.65(dd, **1h**, H-4, 1.2, 7.4); 7.60(dd, **1h**, H-7, 1.8, 7.4); 7.57(td, **1h**, H-5, 7.5, 1.2); 7.47(td, **1h**, H-6, 7.5, 1.8); 7.05(d, 2H, H-11, H-13, 8.6); 6.88(d, 2H, H-10, H-14, 8.6); 5.28(s, 2H, H-8); 2.49(q, 2H, H-15, 7.5); 2.30(s, 3H, H-20′); 1.11(t, 3H, H-15′, 7.5) (Appendix A, for **^1h^**-NMR spectrum).

At 30 °C, the spectrum is composed of broad lines, and we preferred the temperature of 60 °C for simulating the molecular movement.

^13^C-NMR (DMSO-d6, δ ppm, T = 333 K): 177.47(C-16); 169.86(brs, C-1); 156.24(C-9); 149.05(Cq); 148.03(C-21); 138.15(C-19); 136.21(Cq); 135.94(Cq); 133.32(Cq); 131.03(C-5); 130.52(Cq); 128.46(C-7); 128.41(C-4); 128.38(C-11, C-13); 127.66(C-6); 115.09(C-18); 114.70(C-10, C-14); 67.56(C-8); 27.16(C-15); 17.27(C-20′); 15.52(C-15′) (Appendix A, for ^13^C-NMR spectrum).

FT-IR (ATR in solid, ν cm^−1^): 3237s; 3040m; 2963m; 2929m; 2869m; 1661m; 1605w; 1580m; 1526vs; 1471s; 1382m; 1344s; 1291m; 1248m; 1213s; 1149s; 1078w; 1047w; 1014m; 961w; 898w; 823w; 783m; 756m; 724m; 642w; 614w (Appendix A, for IR spectrum).

Chemical Formula: C_23_H_23_N_3_O_2_S, Exact Mass: 405.15110, HRMS (APCI+, DMSO + MeOH): *m*/*z* calcd for [M + H]^+^ C_23_H_24_N_3_O_2_S^+^ = 406.15837, found 406.15724 (100%, ∆m = −2.78 ppm), calcd for C_16_H_18_NO_2_^+^ = 256.13321, found 256.13262 (3%, ∆m = −2.30 ppm), calcd for C_16_H_15_O_2_^+^ = 239.10666, found 239.10710 (10%, ∆m = 1.84 ppm), calcd for C_7_H_7_N_2_S^+^ = 151.03245, found 151.03261 (6%, ∆m = 1.06 ppm).

2-((4-Ethylphenoxy)methyl)-*N*-((5-chloropyridin-2-yl)carbamothioyl)benzamide (**1g**)

(m.p. 135.7–136.9 °C; yield 74%, 3.15 g)

**^1h^**-NMR (DMSO-d6, δ ppm, *J* Hz, T = 303 K): 13.08(s, **1h**, NH, deuterable); 12.13(brs, **1h**, NH, deuterable); 8.76(brs, **1h**, H-18); 8.45(d, **1h**, H-21, 2.4); 8.03(dd, **1h**, H-19, 2.4, 9.0); 7.65(dd, **1h**, H-4, 1.2, 7.4); 7.60(dd, **1h**, H-7, 1.8, 7.4); 7.57(td, **1h**, H-5, 7.5, 1.2); 7.47(td, **1h**, H-6, 7.5, 1.8); 7.03(d, 2H, H-11, H-13, 8.6); 6.86(d, 2H, H-10, H-14, 8.6); 5.27(s, 2H, H-8); 2.47(q, 2H, H-15, 7.5); 1.08(t, 3H, H-15′, 7.5) (Appendix A, for **^1h^**-NMR spectrum).

^13^C-NMR (DMSO-d6, δ ppm, T = 303 K): 177.83(C-16); 170.23(brs, C-1); 156.15(C-9); 148.68(Cq); 146.75(C-21); 137.62(C-19); 136.11(Cq); 135.91(Cq); 133.09(Cq); 131.10(C-5); 130.52(Cq); 128.48(C-7); 128.47(C-11, C-13); 128.46(C-4); 128.35(C-6); 116.38(C-18); 114.49(C-10, C-14); 67.41(C-8); 27.17(C-15); 15.66(C-15′) (Appendix A, for ^13^C-NMR spectrum).

FT-IR (ATR in solid, ν cm^−1^): 3168m; 3027m; 2963m; 2926m; 2863m; 1675m; 1508vs; 1447s; 1379m; 1327s; 1293m; 1246s; 1219s; 1149s; 1106m; 1028m; 956m; 925w; 860m; 767m; 723m; 662m; 624w (Appendix A, for IR spectrum).

Chemical Formula: C_22_H_20_ClN_3_O_2_S, Exact Mass: 425.09648, HRMS (APCI+, DMSO + MeOH): *m*/*z* calcd for [M + H]^+^ C_22_H_21_ClN_3_O_2_S^+^ = 426.10375, found 426.10380 (100%, ∆m = 0.12 ppm), calcd for C_16_H_18_NO_2_^+^ = 256.13321, found 256.13355 (24%, ∆m = 1.33 ppm), calcd for C_16_H_15_O_2_^+^ = 239.10666, found 239.10708 (88%, ∆m = 1.76 ppm), calcd for C_8_H_8_NO^+^ = 134.06004, found 134.06022 (9%, ∆m = 1.34 ppm).

2-((4-Ethylphenoxy)methyl)-*N*-((2-chloropyridin-3-yl)carbamothioyl)benzamide (**1h**)

(m.p. 133.6–134.9 °C; yield 72%, 3.06 g)

**^1h^**-NMR (DMSO-d6, δ ppm, *J* Hz, T= 303 K): 12.52(s, **1h**, NH, deuterable); 12.16(brs, **1h**, NH, deuterable); 8.40(dd, **1h**, H-21, 1.8, 8.0); 8.32(dd, **1h**, H-19, 1.8, 4.7); 7.65(dd, **1h**, H-4, 1.2, 7.4); 7.60(dd, **1h**, H-7, 1.8, 7.4); 7.57(td, **1h**, H-5, 7.5, 1.2); 7.50(dd, **1h**, H-20, 4.7, 8.0); 7.47(td, **1h**, H-6, 7.5, 1.8); 7.08(d, 2H, H-11, H-13, 8.6); 6.90(d, 2H, H-10, H-14, 8.6); 5.28(s, 2H, H-8); 2.51(q, 2H, H-15, 7.5); 1.12(t, 3H, H-15′, 7.5) (Appendix A, for **^1h^**-NMR spectrum).

^13^C-NMR (DMSO-d6, δ ppm, T = 303 K): 180.26(C-16); 170.28(C-1); 156.20(C-9); 147.05(C-19); 145.27(Cq); 136.29(C-21); 136.17(Cq); 135.81(Cq); 133.14(Cq); 132.57(Cq); 131.07(C-5); 128.51(C-4, C-7, C-11, C-13); 127.76(C-6); 122.93(C-20); 114.54(C-10, C-14); 67.52(C-8); 27.21(C-15); 15.75(C-15′) (Appendix A, for ^13^C-NMR spectrum).

FT-IR (ATR in solid, ν cm^−1^): 3098w; 3006m; 2962m; 2931m; 1672m; 1590m; 1516vs; 1451s; 1404m; 1380w; 1325m; 1302m; 1269m; 1245s; 1213m; 1157vs; 1119m; 1064m; 1010m; 951w; 870w; 823w; 755s; 730m; 662w; 640w (Appendix A, for IR spectrum).

Chemical Formula: C_22_H_20_ClN_3_O_2_S, Exact Mass: 425.09648, HRMS (APCI+, DMSO + MeOH): *m*/*z* calcd for [M + H]^+^ C_22_H_21_ClN_3_O_2_S^+^ = 426.10375, found 426.10289 (4%, ∆m = −2.02 ppm), calcd for [M − Cl]^+^ C_22_H_20_N_3_O_2_S^+^ = 390.12707, found 390.12641 (100%, ∆m = −1.69 ppm), calcd for C_17_H_18_NO_2_^+^ = 268.05391, found 268.05347 (7%, ∆m = −1.64 ppm), calcd for C_16_H_18_NO_2_^+^ = 256.13321, found 256.13290 (3%, ∆m = −1.21 ppm), calcd for C_16_H_15_O_2_^+^ = 239.10666, found 239.10637 (13%, ∆m = −1.21 ppm).

2-((4-Ethylphenoxy)methyl)-*N*-((2-chloropyridin-4-yl)carbamothioyl)benzamide (**1i**)

(m.p. 146.1–147.3 °C; yield 65%, 2.76 g)

**^1h^**-NMR (DMSO-d6, δ ppm, *J* Hz, T = 303 K): 12.64(s, **1h**, NH, deuterable); 12.11(brs, **1h**, NH, deuterable); 8.38(d, **1h**, H-20, 5.6); 8.08(d, **1h**, H-18, 1.9); 7.71(dd, **1h**, H-21, 1.9, 5.6); 7.62(dd, **1h**, H-7, 1.8, 7.4); 7.58(dd, **1h**, H-7, 1.2, 7.5); 7.57(td, **1h**, H-5, 7.5, 1.2); 7.47(td, **1h**, H-6, 7.5, 1.8); 7.05(d, 2H, H-11, H-13, 8.6); 6.87(d, 2H, H-10, H-14, 8.6); 5.27(s, 2H, H-8); 2.49(q, 2H, H-15, 7.5); 1.10(t, 3H, H-15′, 7.5) (Appendix A, for **^1h^**-NMR spectrum).

^13^C-NMR (DMSO-d6, δ ppm, T = 303 K): 179.20(C-16); 169.24(C-1); 156.22(C-9); 150.42(C-19); 150.17(C-20); 147.52(Cq); 136.16(Cq); 135.96(Cq); 132.99(Cq); 131.16(C-5); 128.52(C-11, C-13); 128.48(C-4); 128.33(C-7); 127.71(C-6); 116.48(C-21); 116.44(C-18); 114.48(C-10, C-14); 67.41(C-8); 27.19(C-15); 15.69(C-15′) (Appendix A, for ^13^C-NMR spectrum).

FT-IR (ATR in solid, ν cm^−1^): 3334w; 3112w; 2954m; 2926m; 2868w; 1682m; 1574s; 1506vs; 1447m; 1378m; 1320m; 1293m; 1224s; 1145s; 1074w; 1056w; 1022m; 982m; 875w; 860w; 83**1m**; 779w; 730m; 676m (Appendix A, for IR spectrum).

Chemical Formula: C_22_H_20_ClN_3_O_2_S, Exact Mass 425.09648, HRMS (APCI+, DMSO + MeOH): *m*/*z* calcd for [M + H]^+^ C_22_H_21_ClN_3_O_2_S^+^ = 426.10375, found 426.10331 (100%, ∆m = −1.03 ppm), calcd for C_16_H_18_NO_2_^+^ = 256.13321, found 256.13294 (4%, ∆m = −1.05 ppm), calcd for [C_16_H_15_O_2_]^+^ = 239.10666, found 239.10626 (20%, ∆m = −1.67 ppm).

2-((4-Ethylphenoxy)methyl)-*N*-((6-chloropyridin-3-yl)carbamothioyl)benzamide (**1j**)

(m.p. 149.3–150.5 °C; yield 78%, 3.32 g)

**^1h^**-NMR (DMSO-d6, δ ppm, *J* Hz, T=303 K): 12.30(s, **1h**, NH, deuterable); 12.04(brs, **1h**, NH, deuterable); 8.48(d, **1h**, H-21, 2.5); 8.09(dd, **1h**, H-18, 2.5, 8.6); 7.55(d, **1h**, H-19, 8.6); 7.62(dd, **1h**, H-7, 1.8, 7.4); 7.58(dd, **1h**, H-7, 1.2, 7.5); 7.57(td, **1h**, H-5, 7.5, 1.2); 7.47(td, **1h**, H-6, 7.5, 1.8); 7.05(d, 2H, H-11, H-13, 8.6); 6.87(d, 2H, H-10, H-14, 8.6); 5.28(s, 2H, H-8); 2.51(q, 2H, H-15, 7.5); 1.12(t, 3H, H-15′, 7.5) (Appendix A, for **^1h^**-NMR spectrum).

^13^C-NMR (DMSO-d6, δ ppm, T = 303 K): 180.40(C-16); 169.89(C-1); 156.25(C-9); 146.82(C-20); 146.25(C-21); 136.21(C-18); 136.20(Cq); 135.88(C-q); 134.54(Cq); 133.15(Cq); 131.07(C-5); 128.56(C-11, C-13); 128.45(C-4); 128.34(C-7); 127.70(C-6); 123.74(C-19); 114.55(C-10, C-14); 67.44(C-8); 27.22(C-15); 15.76(C-15′) (Appendix A, for ^13^C-NMR spectrum).

FT-IR (ATR in solid, ν cm^−1^): 3170s; 3010m; 2958m; 2928m; 2886m; 1752w; 1691s; 1609w; 1574m; 1513vs; 1451s; 1393m; 1314m; 1275m; 1239s; 1166s; 1098m; 1024m; 824m; 735m; 690m; 652w; 608w (Appendix A, for IR spectrum).

Chemical Formula C_22_H_20_ClN_3_O_2_S, Exact Mass 425.09648, HRMS (APCI+, DMSO + MeOH): *m*/*z* calcd for [M + H]^+^ C_22_H_21_ClN_3_O_2_S^+^ = 426.10375, found 426.10364 (100%, ∆m = −0.26 ppm), calcd for C_16_H_18_NO_2_^+^ = 256.13321, found 256.13373 (3%, ∆m = 2.03 ppm), calcd for C_16_H_15_O_2_^+^ = 239.10666, found 239.10722 (14%, ∆m = 2.34 ppm).

2-((4-Ethylphenoxy)methyl)-*N*-((5-bromopyridin-2-yl)carbamothioyl)benzamide (**1k**)

(m.p. 136.8–138 °C; yield 76%, 3.56 g)

**^1h^**-NMR (DMSO-d6, δ ppm, *J* Hz, T = 303 K): 13.07(brs, **1h**, NH, deuterable); 12.14(brs, **1h**, NH, deuterable); 8.71(brs, **1h**, H-18); 8.53(d, **1h**, H-2.3); 8.14(dd, **1h**, H-19, 2.3, 9.0); 7.64(dd, **1h**, H-4, 1.2, 7.4); 7.60(dd, **1h**, H-7, 1.8, 7.4); 7.57(td, **1h**, H-5, 7.5, 1.2); 7.47(td, **1h**, H-6, 7.5, 1.8); 7.03(d, 2H, H-11, H-13, 8.6); 6.86(d, 2H, H-10, H-14, 8.6); 5.27(s, 2H, H-8); 2.47(q, 2H, H-15, 7.5); 1.09(t, 3H, H-15′, 7.5) (Appendix A, for **^1h^**-NMR spectrum).

^13^C-NMR (DMSO-d6, δ ppm, T = 303 K): 177.81(C-16); 170.22(C-1); 156.15(C-9); 150.02(Cq); 148.94(C-21); 140.39(C-19); 136.12(Cq); 135.92(Cq); 133.09(Cq); 131.11(C-5); 128.49(C-11, C-13); 128.48(C-7); 128.36(C-4); 127.69(C-6); 116.82(C-18); 115.56(C-20); 114.49(C-10, C-14); 67.41(C-8); 27.18(C-15); 15.68(C-15′) (Appendix A, for ^13^C-NMR spectrum).

FT-IR (ATR in solid, ν cm^−1^): 3164m; 3026m; 2961m; 2926m; 2860w; 1674m; 1508vs; 1447s; 1376m; 1326m; 1290m; 1246s; 1220s; 1152s; 1090m; 1027m; 996m; 955w; 926w; 886w; 859w; 824m; 762m; 732s; 658m; 617w (Appendix A, for IR spectrum).

Chemical Formula: C_22_H_20_BrN_3_O_2_S, Exact Mass 469.04596, HRMS (APCI+, DMSO + MeOH): *m*/*z* calcd for [M + H]^+^ C_22_H_21_BrN_3_O_2_S^+^ = 472.05119, found 472.05057 (74%, ∆m = −1.3 ppm), calcd for C_16_H_18_NO_2_^+^ = 256.13321, found 256.13312 (50%, ∆m = −0.35 ppm), calcd for C_16_H_15_O_2_^+^ = 239.10666, found 239.10666 (100%, ∆m = 0 ppm), calcd for C_5_H_6_BrN_2_^+^ = 172.97089, found 172.97096 (6%, ∆m = 0.40 ppm), calcd for C_8_H_8_NO^+^ = 134.06004, found 134.06001 (10%, ∆m = −0.22 ppm), calcd for [Me_2_SO+H]^+^ C_2_H_7_SO^+^ = 79.02121, found 79.02091 (4%, ∆m = −3.80 ppm).

2-((4-Ethylphenoxy)methyl)-*N*-((6-bromopyridin-2-yl)carbamothioyl)benzamide (**1l**)

(m.p. 124.5–125.9 °C; yield 73%, 3.42 g)

**^1h^**-NMR (DMSO-d6, δ ppm, *J* Hz, T = 303 K): 12.97(brs, **1h**, NH, deuterable); 12.16(brs, **1h**, NH, deuterable); 8.69(brs, **1h**, H-18); 7.85(t, **1h**, H-19, 7.9); 7.64(dd, **1h**, H-4, 1.2, 7.4); 7.60(dd, **1h**, H-7, 1.8, 7.4); 7.57(td, **1h**, H-5, 7.5, 1.2); 7.51(d, **1h**, H-20, 7.9); 7.47(td, **1h**, H-6, 7.5, 1.8); 7.03(d, 2H, H-11, H-13, 8.6); 6.86(d, 2H, H-10, H-14, 8.6); 5.28(s, 2H, H-8); 2.47(q, 2H, H-15, 7.5); 1.09(t, 3H, H-15′, 7.5) (Appendix A, for **^1h^**-NMR spectrum).

^13^C-NMR (DMSO-d6, δ ppm, T = 303 K): 177.98(C-16); 170.11(C-1); 156.16(C-9); 151.18(Cq); 141.03(C-19); 138.73(Cq); 136.13(Cq); 135.97(Cq); 133.02(Cq); 131.01(C-5); 128.53(C-7); 128.50(C-11, C-13); 128.30(C-4); 127.68(C-6); 125.07(C-20); 114.55(C-10, C-14); 114.38(C-18); 67.40(C-8); 27.19(C-15); 15.70(C-15′) (Appendix A, for ^13^C-NMR spectrum).

FT-IR (ATR in solid, ν cm^−1^): 3210m; 3172m; 3026m; 2971m; 29286m; 2872w; 1672m; 1556s; 1509vs; 1451m; 1421s; 1322s; 1255m; 1223s; 1170s; 1144m; 1120s; 1069w; 1026m; 980w; 820m; 762m; 725m; 660m (Appendix A, for IR spectrum).

Chemical Formula: C_22_H_20_BrN_3_O_2_S, Exact Mass 469.04596, HRMS (APCI+, DMSO + MeOH): *m*/*z* calcd for [M + H]^+^ C_22_H_21_BrN_3_O_2_S^+^ = 472.05119, found 472.04990 (100%, ∆m = −2.73 ppm), calcd for C_16_H_18_NO_2_^+^ = 256.13321, found 256.13287 (4%, ∆m = −1.33 ppm), calcd for C_16_H_15_O_2_^+^ = 239.10666, found 239.10634 (63%, ∆m = −1.34 ppm).

2-((4-Ethylphenoxy)methyl)-*N*-((3,5-dichloropyridin-2-yl)carbamothioyl)benzamide (**1m**)

(m.p. 125.1–126.5 °C; yield 64%, 2.94 g)

**^1h^**-NMR (DMSO-d6, δ ppm, *J* Hz, T = 303 K): 12.23(s, **1h**, NH, deuterable); 12.09(s, **1h**, NH, deuterable); 8.55(d, **1h**, H-21, 2.3); 8.35(d, **1h**, H-19, 2.3); 7.65(dd, **1h**, H-4, 1.2, 7.4); 7.60(dd, **1h**, H-7, 1.8, 7.4); 7.57(td, **1h**, H-5, 7.5, 1.2); 7.47(td, **1h**, H-6, 7.5, 1.8); 7.09(d, 2H, H-11, H-13, 8.6); 6.90(d, 2H, H-10, H-14, 8.6); 5.26(s, 2H, H-8); 2.52(q, 2H, H-15, 7.5); 1.14(t, 3H, H-15′, 7.5) (Appendix A, for **^1h^**-NMR spectrum).

^13^C-NMR (DMSO-d6, δ ppm, T = 303 K): 180.54(C-16); 169.84(C-1); 156.18(C-9); 147.43(Cq); 145.62(C-21); 137.97(C-19); 136.18(Cq); 135.84(Cq); 133.07(Cq); 131.11(C-5); 129.95(Cq); 129.00(Cq); 128.59(C-7); 128.56(C-11, C-13); 128.55(C-4); 127.74(C-6); 114.70(C-10, C-14); 67.45(C-8); 27.25(C-15); 15.79(C-15′) (Appendix A, for ^13^C-NMR spectrum).

FT-IR (ATR in solid, ν cm^−1^): 3129m; 3027m; 2962m; 2927m; 2868w; 1671m; 1560m; 1504vs; 1438m; 1231s; 1151s; 1114s; 1050m; 953w; 895w; 853w; 823m; 777m; 743w; 708w; 659w; 610w (Appendix A, for IR spectrum).

Chemical Formula: C_22_H_19_Cl_2_N_3_O_2_S, Exact Mass 459.05750, HRMS (APCI+, DMSO + MeOH): *m*/*z* calcd for [M + H]^+^ = C_22_H_20_Cl_2_N_3_O_2_S^+^ 460.06478, found 460.06464 (20%, ∆m = −0.30 ppm), calcd for C_16_H_18_NO_2_^+^ = 256.13321, found 256.13330 (12%, ∆m = 0.35 ppm), calcd for C_16_H_15_O_2_^+^ = 239.10666, found 239.10677 (100%, ∆m = 0.46 ppm), calcd for C_8_H_8_NO^+^ = 134.06004, found 134.06010 (13%, ∆m = 0.45 ppm).

2-((4-Ethylphenoxy)methyl)-*N*-((2,6-dichloropyridin-4-yl)carbamothioyl)benzamide (**1n**)

(m.p. 138.6–140 °C; yield 61%, 2.80 g)

**^1h^**-NMR (DMSO-d6, δ ppm, *J* Hz, T = 303 K): 12.66(s, **1h**, NH, deuterable); 12.19(s, **1h**, NH, deuterable); 8.01(s, 2H, H-18, H-21); 7.65(dd, **1h**, H-4, 1.2, 7.4); 7.60(dd, **1h**, H-7, 1.8, 7.4); 7.57(td, **1h**, H-5, 7.5, 1.2); 7.47(td, **1h**, H-6, 7.5, 1.8); 7.05(d, 2H, H-11, H-13, 8.6); 6.87(d, 2H, H-10, H-14, 8.6); 5.26(s, 2H, H-8); 2.50(q, 2H, H-15, 7.5); 1.10(t, 3H, H-15′, 7.5) (Appendix A, for **^1h^**-NMR spectrum).

^13^C-NMR (DMSO-d6, δ ppm, T = 303 K): 179.38(C-16); 169.85(C-1); 156.21(C-9); 149.60(Cq); 149.23(C-20, C-19); 136.16(Cq); 135.99(Cq); 132.910(Cq); 131.22(C-5); 128.50(C-7); 128.49(C-11, C-13); 128.36(C-4); 127.74(C-6); 116.01(C-18, C-21); 114.45(C-10, C-14); 67.39(C-8); 27.18(C-15); 15.68(C-15′) (Appendix A, for ^13^C-NMR spectrum).

FT-IR (ATR in solid, ν cm^−1^): 3279m; 3116m; 2958w; 2927m; 2869m; 1681m; 1574s; 1503vs; 1372m; 1296s; 1232s; 1163s; 1123m; 1097m; 1043m; 1002w; 884w; 846w; 820m; 773w; 738w; 678m; 612w (Appendix A, for IR spectrum).

Chemical Formula: C_22_H_19_Cl_2_N_3_O_2_S, Exact Mass 459.05750, HRMS (APCI+, DMSO + MeOH): *m*/*z* calcd for [M + H]^+^ C_22_H_20_Cl_2_N_3_O_2_S^+^ = 460.06478, found 460.06400 (100%, ∆m = −1.70 ppm), calcd for [M − S + OH]^+^ C_22_H_20_Cl_2_N_3_O_3_^+^ = 444.08762, found 444.08749 (17%, ∆m = −0.29 ppm), calcd for C_16_H_18_NO_2_^+^ = 256.13321, found 256.13300 (21%, ∆m = −0.82 ppm), calcd for C_16_H_15_O_2_^+^ = 239.10666, found 239.10649 (43%, ∆m = −0.71 ppm), calcd for C_5_H_5_Cl_2_N_2_^+^ = 162.98243, found 162.98230 (5%, ∆m = −0.80 ppm), calcd for C_8_H_8_NO^+^ = 134.06004, found 134.05984 (5%, ∆m = −1.49 ppm), calcd for [Me_2_SO+H]^+^ C_2_H_7_SO^+^ = 79.02121, found 79.02081 (8%, ∆m = −5.06 ppm).

2-((4-Ethylphenoxy)methyl)-*N*-((3,5-dibromopyridin-2-yl)carbamothioyl)benzamide (**1o**)

(m.p. 140.2–142.9 °C; yield 54%, 2.95 g)

**^1h^**-NMR (DMSO-d6, δ ppm, *J* Hz, T = 303 K): 12.24(s, **1h**, NH, deuterable); 12.09(s, **1h**, NH, deuterable); 8.65(d, **1h**, H-21, 2.1); 8.54(d, **1h**, H-19, 2.1); 7.63(dd, **1h**, H-4, 1.2, 7.5); 7.61(dd, **1h**, H-7, 1.8, 7.5); 7.57(td, **1h**, H-5, 7.5, 1.2); 7.48(td, **1h**, H-6, 7.4, 1.8); 7.10(d, 2H, H-11, H-13, 8.6); 6.91(d, 2H, H-10, H-14, 8.6); 5.27(s, 2H, H-8); 2.53(q, 2H, H-15, 7.5); 1.14(t, 3H, H-15′, 7.5) (Appendix A, for **^1h^**-NMR spectrum).

^13^C-NMR (DMSO-d6, δ ppm, T = 303 K): 180.38(C-16); 169.81(C-1); 156.18(C-9); 148.98(C-17); 148.34(C-21); 143.47(C-19); 136.16(Cq); 135.85(Cq); 133.04(Cq); 131.13(C-5); 128.62(C-7); 128.57(C-11, C-13); 127.75(C-6); 119.54(C-20); 118.52(C-18); 114.70(C-10, C-14); 67.39(C-8); 27.25(C-15); 15.79(C-15′) (Appendix A, for ^13^C-NMR spectrum).

FT-IR (ATR in solid, ν cm^−1^): 3130m; 3027w; 2965w; 2844w; 1673m; 1546m; 1504vs; 1430m; 1360m; 1323w; 1236s; 1150vs; 1043m; 952w; 825w; 737m; 696w; 661w (Appendix A, for IR spectrum).

Chemical Formula: C_22_H_19_Br_2_N_3_O_2_S, Exact Mass 546.95647, HRMS (APCI+, DMSO + MeOH): *m*/*z* calcd for [M + H]^+^ C_22_H_20_Br_2_N_3_O_2_S^+^ = 549.96170, found 549.96069 (100%, ∆m = −1.84 ppm), calcd for C_14_H_9_BrN_3_OS^+^ = 345.96442, found 345.96356 (8%, ∆m = −2.49 ppm), calcd for C_16_H_18_NO_2_^+^ = 256.13321, found 256.13354 (20%, ∆m = 1.29 ppm), calcd for C_16_H_15_O_2_^+^ = 239.10666, found 239.10707 (99%, ∆m = 7.71 ppm), calcd for C_8_H_8_NO^+^ = 134.06004, found 134.06015 (10%, ∆m = 0.82 ppm), calcd for [Me_2_SO + H]^+^ C_2_H_7_SO^+^ = 79.02121, found 79.02081 (15%, ∆m = −5.06 ppm).

### 2.3. Biological Evaluation of the Antimicrobial Activity

#### 2.3.1. Quantitative Evaluation of the Antimicrobial Activity

The in vitro evaluation of the antimicrobial activity of the compounds was performed using the following standard microbial strains: *Staphylococcus aureus* ATCC 25923, *Enterococcus faecalis* ATCC 29212, *Escherichia coli* ATCC 25922, and *Pseudomonas aeruginosa* ATCC 27853. Fresh microbial cultures grown for 18–24 h on *Plate Count Agar* were used to prepare suspensions in sterile PBS (Phosphate-Buffered Saline). The 0.5 McFarland standard was used as a reference to adjust the turbidity of the microbial suspensions to 1.5 × 10^8^ colony-forming units (CFU/mL). The compounds were solubilized in DMSO at a stock concentration of 10 mg/mL. Then, serial binary dilutions were prepared in liquid culture medium (Mueller–Hinton) in 96-well plates, starting from 5 mg/mL (the compound testing concentrations were between 5 and 0.009 mg/mL). The medium containing different concentrations of the compounds was further inoculated with the microbial suspensions at a final density of 10^6^ CFU/mL. The inoculated 96 well-plates were incubated for 18–24 h at 37 °C. The positive control was represented by ciprofloxacin (5 µg/mL). Growth control was represented by the broth inoculated only with standard microbial suspension. The determinations were performed in triplicate to confirm the MIC (minimum inhibitory concentration) values. The MIC values were determined by visual inspection, as the lowest concentrations of tested compounds that inhibited the microbial growth in the liquid medium (the culture medium remained clear, similar to the sterility control).

#### 2.3.2. Evaluation of the Antibiofilm Activity

The compounds’ antibiofilm activity was investigated using the crystal violet microtiter assay. After the determination of the MIC, the 96-well plates were emptied and washed three times with sterile saline to remove non-adherent microbial cells. Microbial cells adhering to the walls of the wells were fixed with 150 µL methanol for 5 min, then stained with 150 µL 1% violet crystal solution (prepared in distilled water) for 20 min. The dye was removed, and the plates were washed using sterile saline. The microbial biofilms formed on the inert substrate were suspended in 33% acetic acid, then the absorbance was read at a wavelength of 490 nm. The minimum inhibitory concentration of the total biofilm mass development (MBIC) was defined as the lowest concentration of compound that induced a decrease in the stained biofilm mass and, consequently, of the absorbance value, at levels similar to those of the sterility control.

### 2.4. Total Antioxidant Activity (TAC)

A solution of 2,2-diphenyl-1-picrylhydrazyl (DPPH) with a concentration of 4 × 10^−4^ M in acetone was prepared. Ascorbic acid was used as a reference. Stock solutions of ascorbic acid and compounds **1a**–**1o** were prepared in acetone at a standard concentration of 2 mg/mL. For TAC measurements, 1 mL of stock solution of DPPH was added to 1 mL of stock solution of each compound, and the mixture was kept for 30 min in the dark, followed by the absorbance measuring at 517 nm. A UVD-3500 UV-Vis spectrophotometer was used for this purpose. The TAC percentage was calculated by Equation (1):(1)$$TAC,\%=\frac{Abs_{i}-Abs_{30min}}{Abs_{i}}\times 100$$
where Abs_i_ is the initial absorbance of the mixture, and the Abs_30 min_ is the absorbance measured after 30 min [26,27,28].

### 2.5. RP—HPLC Analytical Method

#### 2.5.1. Materials and Equipment

HPLC grade solvents were used in the HPLC analysis: acetonitrile reagent (Fischer Scientific, Waltham, MA, USA), methanol reagent (Scharlau, Hamburg Germany), triethylamine (Sigma-Aldrich, Munich, Germany), and orthophosphoric acid (Fisher Scientific, Waltham, MA, USA).

The Waters Alliance HPLC system, comprised of the following modules: 2695 + 2998 separation module, 998 PDA detector, and PC equipped with “Empower no. 3 PDA Software”, was used for the determination. The samples were weighed by using a Mettler Toledo analytical balance, and the pH of the solutions was determined with a pH meter inoLab pH7310P. To filter the solutions injected in the HPLC system, syringe nylon filters (0.45 µm) (Agilent Technologies, Santa Clara, CA, USA) were used.

#### 2.5.2. Chromatographic Conditions

The RP-HPLC analytical method was developed and optimized by using a chromatographic column with a stationary phase composed of octadecyl silica gel (C18), 250 mm length, 4.6 mm diameter, and particle size of 5 μm (Inertsil ODS-3, 5 μm, 250 × 4.6 mm). The chosen column and the samples were maintained at room temperature during the determinations.

The mobile phase, represented by a solvent mixture of pH 3.3 and water (93:7, *v*/*v*), flowed through the chromatographic system with a flow rate of 1.0 mL/min, in isocratic elution. A total of 10 μL of the sample to be separated was injected into the system, the run time lasted for 35 min per registered chromatogram, and the detection of the separated components was measured at 275 nm wavelength.

#### 2.5.3. Mobile Phase Preparation

The mobile phase was comprised of 93% solvent A and 7% water R (of chromatographic use) (93:7, *v*/*v*), where solvent A was prepared by mixing water R and acetonitrile R (20 80, *v*/*v*), 0.7 mL triethylamine R added, and the pH adjusted to 3.3 with ortho-phosphoric acid R. The obtained solution was filtered and sonicated for around one hour before use.

#### 2.5.4. Samples Preparation

The compounds **1h**, **1j**, **1i**, **1m**, **1g**, and **1n** were intended to be separated from the mixture. Challenges regarding their close chemically related structures, their purity, and difficulty in eluting one component at a different time point were considered and the current analytical method was developed to overcome the issue.

##### System Suitability

A test solution that contained 5 µg/mL from each of the components was injected into the chromatographic system, following the optimized chromatographic conditions, and the retention time of the main peaks due to tested compounds, resolution, theoretical plates, and symmetry factors were evaluated.

Individual stock solutions of 500 µg/mL were prepared from every tested chemical substance (**1h**, **1j**, **1i**, **1m**, **1g**, and **1n**). An amount of 5 mg was transferred into a 10 mL volumetric flask, 5 mL acetonitrile R was added, and the flask was sonicated for 5 min, cooled to room temperature, and diluted to volume with acetonitrile R. The final test solution was obtained by diluting 0.25 mL from each stock solution to 25 mL with methanol R, leading to a concentration of 5 µg/mL from each of the compounds **1h**, **1j**, **1i**, **1m**, **1g**, and **1n**.

##### Specificity

To evaluate the specificity of the method, individual solutions of 5 µg/mL from each chemical entity were injected, along with the diluent. The identification solutions were prepared firstly by obtaining stock solutions of 500 µg/mL in acetonitrile R, followed by their dilution of 0.25 mL to 25 mL with methanol R, to reach the concentration of 5 µg/mL.

The interference of diluent in the identification of compounds **1h**, **1j**, **1i**, **1m**, **1g**, and **1n** was evaluated.

##### LOD—LOQ

The limits of detection (LOD) and quantification (LOQ) address the ability of the method to detect and quantify low amounts of analytes in the samples. There are different approaches for LOD and LOQ calculation; currently, the parameters are determined statistically based on the calibration curve, using standard deviation (σ) and slope (S), as expressed in Equations (2) and (3) [29].
(2)$$LOD=\frac{3.3\times \sigma }{S}$$
(3)$$LOD=\frac{10\times \sigma }{S}$$

Increasingly concentrated solutions in the range of 0.06–0.5 µg/mL were prepared for each compound. Firstly, individual stock solutions were prepared from each solid dissolved in acetonitrile R, in a suitable amount to reach 500 µg/mL concentration. Furthermore, a cumulative stock solution of 4.0 µg/mL was obtained by diluting each stock solution to a specific volume with methanol R. The desirable number of concentrations was obtained by dilution of cumulative stock solution (the detailed preparation of concentration ranges is included in the Appendix A—RP-HPLC Analytical Report and Chromatograms, LOD and LOQ chapters).

The calibration plot that sustained the linear relationship between the area under registered peaks and their concentrations was traced. A linear calibration curve is described by a regression model, a correlation coefficient (R^2^ ≥ 0.99), standard deviation, and slope of the line.

##### Linearity

As per Q2(R1) [30], the linearity represents the capacity to obtain results directly proportional to the concentration of the analyte in the prepared sample solution.

A linear calibration curve is a positive indication of analytical method performance in a specific concentration range.

The linear relationship between the concentration (X-axis) of the analyte and the response (Y-axis) was plotted. The obtained calibration curve was evaluated in terms of correlation coefficient (R^2^ ≥ 0.99), slope, and y-intercept.

For the current analytical method, the linearity was studied from the reporting level of each chemical to 120% of the established concentration. Specifically, the method has been tested for linearity in a range of theoretical working concentrations: LOQ—6 µg/mL (0.06 µg/mL, 0.08 µg/mL, 0.1 µg/mL, 0.2 µg/mL, 0.5 µg/mL, 4.0 µg/mL, 5.0 µg/mL, 6 µg/mL). The successive dilutions were prepared from stock solutions of 500 µg/mL, using acetonitrile R as solvent. The desired concentrations were obtained by suitable dilutions in methanol R (the detailed preparation of concentration ranges is included in the Appendix A—RP-HPLC Analytical Report and Chromatograms, Linearity chapter).

##### Precision

The precision of the analytical method has been demonstrated by intra-assay and intermediate precision.

Six solutions of 5.0 µg/mL **1h**, **1j**, **1i**, **1m**, **1g**, and **1n** each were prepared, injected in the chromatographic system, and registered on two consecutive days, then evaluated for relative standard deviation. The solutions were prepared from stock solutions of 500 µg/mL, with acetonitrile R used as a solvent. To reach 5.0 µg/mL concentration, dilutions in methanol R were realized (extensive preparation of the solutions is included in the Appendix A—RP-HPLC Analytical Report and Chromatograms, Precision chapter).

##### Accuracy

An accuracy study was developed by preparing increasingly concentrated solutions of analytes in the range of 4–6 µg/mL (considering 4 µg/mL at the level of 80%, 5 µg/mL–100%, 6 µg/mL–120%). For each concentration level, six solutions were prepared and analyzed (extensive preparation of the solutions can be consulted in the Appendix A—RP-HPLC Analytical Report and Chromatograms, Accuracy chapter). The determined and theoretical concentrations were calculated and the recovery was determined, along with the confidence interval.

##### Robustness

The parameters altered to demonstrate the robustness of the method were chosen concerning the percentage of mobile phase (solvent A: solvent B, 91:9, *v*/*v*, and solvent A: solvent B, 95:5, *v*/*v*), flow rate (±0.2 mL/ min), and column temperature (35 °C).

The variations were made from the validated chromatographic conditions: mobile phase = Solvent A: Solvent B (93:7, *v*/*v*), flow rate = 1.0 mL/min, column temperature = room temperature.

The robustness was checked based on a solution of 5.0 µg/mL **1h**, **1j**, **1i**, **1m**, **1g**, and **1n** each, prepared according to the RP-HPLC Analytical Report and Chromatograms, Robustness chapter, enclosed in the Appendix A.

The effect of variations has been examined in terms of the retention time of the main peaks, and resolution.

## 3. Results

### 3.1. Chemistry

The condensation between different acid chlorides with ammonium thiocyanate and the reaction of isothiocyanates as key intermediates obtained in situ, with amines, is one of the most frequently used synthetic routes to obtain the *N*-acyl thiourea derivatives.

Thus, the new compounds (**1a**–**1o**) resulted from treating 2-((4-ethylphenoxy)methyl)benzoyl isothiocyanate (2) with a heterocyclic amine. The isothiocyanate was obtained in the reaction of 2-((4-ethylphenoxy)methyl)benzoic acid chloride (3) with ammonium thiocyanate.

The acid chloride (3) was prepared by refluxing 2-((4-ethylphenoxy)methyl)benzoic acid (4) with thionyl chloride in a dichloroethane medium. The acid (4) resulted from acidification with a mineral acid of the corresponding potassium salt (5), which in turn resulted from phthalide (6) and 4-ethylphenol.

The preparation of the new derivatives is described in Figure 1.

### 3.2. Spectral Data

Spectral methods confirmed the chemical structure of the new compounds: Fourier transform infrared (FT-IR), nuclear magnetic resonance (NMR), and atmospheric pressure chemical ionization (APCI) mass spectrometry (MS).

In the **^1h^**-NMR spectra, the -NH protons resonated as singlets or broad singlets in the range of 12.11–13.08 ppm and 11.82–12.60 ppm. The -CH2-O- protons signal was observed in the region 5.26–5.28 as a singlet. The ethyl group protons appear as a triplet at 1.08–1.14 ppm for -CH3 group, and as a quartet at 2.47–2.53 ppm for -CH2- group.

In the ^13^C-NMR, the C=O and C=S carbons gave signals in the regions 166.72–170.28 ppm, and 177.47–180.54 ppm, respectively. The methylene carbon of -CH2-O group appears in the range of 67.29-67.56 ppm and the ethyl group carbons appear at 15.52–15.79 ppm (-CH3) and 27.16–27.25 ppm (-CH2-).

APCI+ high-resolution mass spectra were recorded for all compounds in a mixture of DMSO and MeOH. The molecular peaks [M + H]^+^ were observed as base peaks for ten compounds out of fifteen (**1b**, **1c**, **1e**, **1f**, **1g**, **1i**, **1j**, **1l**, **1n**, and **1o**), thus confirming the identity of the investigated species. For **1h**, the base peak is corresponding to the [M − Cl]^+^ cation. In the mass spectra of compounds **1a**, **1d**, **1k**, and **1m**, the base peak is related to the [C_16_H_15_O_2_]^+^ fragment (calculated *m*/*z* 239.10666), whose structure is depicted on the scheme below (Figure 1). Other frequently observed peaks in the mass spectra correspond to the [C_16_H_18_NO_2_]^+^ and [C_8_H_8_NO]^+^ cations at calculated *m*/*z* values of 256.13321 and 134.06004, whose structures are drawn in Figure 1.

### 3.3. Biological Evaluation of the Antimicrobial Activity

The results of the quantitative evaluation of the antimicrobial activity of the analyzed compounds are presented in Table 1. The MIC values, determined against the two Gram-negative bacteria: *E. coli* ATCC 25922 and *P. aeruginosa* ATCC 27853, and Gram-positive bacteria: *S. aureus* ATCC 25923 and *E. faecalis* ATCC 29212, ranged from 2500 to 625 µg/mL, indicating that the compounds exhibited low antimicrobial activity, as compared to the positive control represented by ciprofloxacin (MICs values of 0.012–0.62 µg/mL).

Regarding the influence of the obtained compounds on bacterial adherence, the results showed a low antibiofilm activity, by comparison with the antibiotic control, with MBICs values of >5000–312 µg/mL (results attached in Table 2).

### 3.4. Total Antioxidant Capacity Measurements

The total antioxidant capacity (TAC) was measured using the well-known DPPH method, as mentioned previously. Ascorbic acid was used as a reference, with a determined antioxidant capacity of 100%. The obtained results are shown in Table 3. The highest antioxidant capacity was recorded for the compound **1i** (87%), followed by **1a** (44%), while for the other compounds, TAC values were between 0 and 29%. The TAC value of every compound was calculated with regard to the antioxidant capacity of the reference (ascorbic acid, TAC = 100%).

### 3.5. Analytical Method Validation by PR-HPLC Technique

#### 3.5.1. Specificity

The system suitability of the analytical method was inspected by monitoring the asymmetry of chromatographic peaks, their reasonable separation, along with the purity and theoretical plates. Methanol as a diluent was used for sample preparation and individual solutions of 5 µg/mL compounds **1h**, **1j**, **1i**, **1m**, **1g**, and **1n** were injected in the chromatographic system. The retention time and UV spectra were extracted for peak identification (Figure 2).

A solution that contained 5 µg/mL of each of the related compounds (**1h**, **1j**, **1i**, **1m**, **1g**, and **1n**) was injected, the chromatogram registered (Figure 3), and the peaks evaluated for system suitability parameters (the checked chromatogram was extracted from linearity validation parameter, solution 5 µg/mL **1h**, **1j**, **1i**, **1m**, **1g**, and **1n**).

For every compound revealed on the chromatogram, the symmetry factor (tailing factor) was in the range of 1.02–1.07, confirming the Gaussian symmetry of the peaks [31]. The degree of separation was checked through resolution; the chromatographic peaks were fully separated, as the resolution between two adjacent peaks proved to be greater than 2 [32]. Moreover, as the purity angle (PA) had values lower than the purity threshold (TH), the peak purity was confirmed. The theoretical plates (N) are a measure of column efficacy [33]; for every peak, the plate count was more than 11,000 (Table 4).

The interference study aims to demonstrate the lack of interference between the solvent used at solution preparation and the main peaks on the chromatogram. Figure 4 and Figure 5 are typical HPLC chromatograms that demonstrate no interference between diluent and eluted compounds.

The results sustain the hypothesis that the optimized analytical RP—HPLC method is well suited for the proper separation of the chemically related compounds **1h**, **1j**, **1i**, **1m**, **1g**, and **1n**.

#### 3.5.2. LOD and LOQ

The calibration curve was traced by preparing increasingly concentrated solutions for every related compound, in the range of 0.06 µg/mL–0.5 µg/mL.

As mentioned, the calibration curve was described by a linear regression equation, where a linear relationship between the areas under the curve and their corresponding concentrations was proved by a correlation coefficient (R^2^) greater than 0.99, for each series of solutions (the calibration plots for the related compounds are attached in the Appendix A—HPLC Reports, along with the extensive description of solutions’ preparation). LOD and LOQ values were estimated by regression analysis, considering the standard deviation and slope, calculated for every plot (Table 5, Figure 6).

For the investigated compounds, the limit of detection was set in the range 0.0184 µg/mL–0.0252 µg/mL, while the limit of quantification was within 0.0614 µg/mL–0.0797 µg/mL. The method based on the dispersion characteristics of the regression line [34,35] was able to point out the lowest distinguishable and the lowest quantifiable concentrations of the chemicals.

#### 3.5.3. Linearity

For the linearity parameter, the regression line was traced in the concentration range of 0.06–6.00 µg/mL. The linear relationship between concentration and response was demonstrated for every compound, by a correlation coefficient (R^2^) greater than 0.99 (Table 6, Figure 7).

#### 3.5.4. Precision Results

Precision is defined as the degree of proximity among a set of results.

Six samples were prepared to determine the intra-assay precision (Table 7 and Table 8), while for the intermediate precision that measured the precision of the analytical method, six samples were prepared on two different days (Table 9). The precision of the results was evaluated by the relative standard deviation (RSD) values in the series of measurements. RSD values < 2.0% were recorded in every scenario, proving the precision of the analytical method.

#### 3.5.5. Accuracy Results

The accuracy of the analytical method was assessed by determining the degree of closeness between the real (theoretical) values of the analytes in the sample solutions and their determined concentrations. The accuracy parameter was performed for every compound, over three concentration levels, six replicates each (4 µg/mL–80%, 5 µg/mL–100%, 6 µg/mL–120%). Consequently, the accuracy was expressed as the percentage recovery of the determined amount of each analyte in the sample solutions, along with the relative standard deviations and confidence interval (Table 10 is an example of the calculus carried out to determine the recovery of the analyte **1i** at each level of concentration. For every tested compound, the accuracy results are filled in the Appendix A, RP-HPLC Analytical Report and Chromatograms, Appendix A). The recovery of each compound was determined, with results in a 95% confidence level. RSDs on the obtained results were in the range of 0.62–1.33%, meaning that the obtained recovery values were uniformly and tightly distributed around the mean.

#### 3.5.6. Robustness Results

The robustness of the chromatographic method was evaluated by inducing deliberated variations in the method parameters. To provide evidence of the reliability of the method, alterations concerning mobile phase percentage, flow rate and column temperature were made and evaluated for a solution of 5 µg/mL **1h**, **1j**, **1i**, **1m**, **1g**, **1n** (Table 11). The resolution as a system suitability control parameter was checked in every tested scenario. As the resolution values proved to be more than 2, it was generally considered that the chromatographic peaks were completely separated and the developed analytical method is reliable and robust under the selected conditions.

## 4. Discussion

The current research paper is a continuation of our previous efforts in this area [24], synchronizing with the current needs of novel antimicrobials to overcome the increasing resistance to accessible antibiotics. In this context, the study aimed to synthesize new *N*-acyl thiourea derivatives (referred to in the current paper as **1a**–**1o**), that incorporate both thiazole or pyridine nucleus in the same molecule. The present research is built on our previous in silico prediction and molecular docking studies, suggesting the antimicrobial potential of the designed compounds [24]. First, a series of molecular descriptors (area, volume, polar surface area, ovality, log P, polarizability, the energy of the frontier molecular orbitals) have been calculated based on the chemical structure of **1a**–**1o**, through in silico computational approaches, to predict their drug-like, absorption, distribution, metabolism, elimination, and toxicity (ADMET). According to Lipinski’s rule, a promising chemical candidate with oral absorption and adequate permeability is characterized by a molecular mass not more than 500 Da, having maximum 10 H-bond acceptors, 5 H-bond donors and partition coefficient (log P) of 5 [36,37,38]. In the evaluated series, five compounds respected the Lipinski’s rule of five (**1a**, **1b**, **1c**, **1e**, and **1f**), **1o** showed two violations (log P > 5, molecular mass > 500 Da), and the others showed one violation from the rule (log P > 5).

The biological activity of the studied compounds **1a**–**1o** has been previously checked through molecular docking studies [24]. The most favorable conformation of the ligands in the active site of receptor proteins (*S. aureus* DNA gyrase B and *E. coli* DNA gyrase B), has been confirmed by the presence of *H*-bonds established with the amino acid scaffolds of the selected receptors, and satisfactory docking scores. Consequently, the results obtained after generating the in silico studies provided an impetus to synthesize and evaluate the in vitro biological activities of chemicals **1a**–**1o**, on bacterial strains which are among the most threatening and challenging because of their resistance to all currently available antibiotics and high biofilm development ability. Among Gram-positive bacteria, *S. aureus* and *E. faecalis* are two of the most frequent Gram-positive opportunistic and nosocomial pathogens, exhibiting resistance to penicillin, methicillin, quinolones, aminoglycosides, macrolides, glycopeptides, and often harboring multi-drug resistance (MDR) phenotypes [39]. The Gram-negative bacilli, both glucose-fermentative (*E. coli*) and non-fermentative (*P. aeruginosa*), are involved in endogenous and exogenous infections that are very difficult to treat due to their multiple intrinsic and acquired resistance sometimes to the majority (extended-drug resistance) or even all tested antibiotics (pan-drug resistance), including last resorts such as plasmid-mediated colistin resistance [40,41]. The resistance determinants are often located on mobile genetic elements facilitating their escape from clinical settings to the community and the environment [42,43]. In addition to their genetic resistance, these opportunistic pathogens have also the ability to develop sessile communities on different tissues or implanted devices, and are very tolerant to high antibiotic concentrations. They are impossible to reach in vivo, thus inducing biofilm-associated infections, which are often hard to treat and lead to chronic, persistent conditions [44].

Among the tested compounds, **1a**, **1g**, **1h**, and **1o** had better activity in the series against *S. aureus* ATCC 25923, *P. aeruginosa* ATCC 27853 with moderate MIC values of 625 µg/mL.

The compound **1a** was designed with the thiazole ring on the backbone molecule. Structurally, thiazole has an electron-donating group (-S-) and an electron-accepting group (-C=N-). The thiazole nucleus demonstrated its activity against multi-drug resistant Gram-negative bacteria (*P. aeruginosa*) in recent studies [45,46].

The analogues substituted with halogens in the pyridine group showed similar antimicrobial activity against the tested bacterial strains. The derivatives designed with chlorine (**1g**: *5-chloro*; **1h**: *2-chloro*) and bromine (**1o**: *3,5-dibromo*) atoms on the pyridine ring distinguished in the series through higher activity. Thus, it may be inferred that the position of the halogen substituents improved the activity of the resulting compounds, in comparison with other designed isomers.

The moderate or low inhibition of the compounds may be the result of unsatisfactory penetration of the bacterial membrane. High log P results and/or the molecular mass can affect the bacterial membrane crossing [47]. Establishing a successful structure–activity relationship and the optimization of compound characteristics through the synthesis and analysis of a wider number of compounds remain focuses of compound design improvement that need to be addressed in the future, to induce better antibacterial activity.

From the tested compounds, **1e** exhibited the best antibiofilm activity against *E. coli* ATCC 25922, with an MBIC value of 312 µg/mL.

Remarkably, the compounds containing chlorine atoms in the molecule are designed as position isomers (i.e., **1g**, **1h**, **1i**, and **1j**; **1n** with **1m**). The relationship among compounds with respect to the chemical structure and their molecular properties inspired us to evaluate their similar chromatographic characteristics via the HPLC technique. In such cases of related substances, the phenomenon of peak co-elution becomes challenging. As the peak overlapping represents an issue in chromatography, in the current study, an RP-HPLC method is proposed and optimized to simultaneously separate and quantify the chemically related compounds **1g**, **1h**, **1i**, **1j**, **1n**, **1m**.

For the quantification of *N*-acyl-thiourea derivatives, we developed and reported a new RP-HPLC analytical method in a previous study [48]. The analysis was completed with an isocratic elution of 80% solvent A and 20% solvent B (*v*/*v*). Solvent A was represented by 6.8 g/L potassium dihydrogen phosphate R, while acetonitrile R, HPLC grade, represented solvent B. The method has also been validated and proved to be specific, precise, accurate, and linear in a particular range of concentrations, confirming that it is suited for the determination and quantification of 2-((4-methoxyphenoxy)methyl)-*N*-((6-methylpyridin-2-yl)carbamothioyl)benzamide.

Evaluating the data, both methods give similar results in terms of recovery (%), precision, and linearity. Despite these aspects, the newly developed and optimized method (as compared against the previously mentioned method) as well the increased simplicity of the experimental setup, i.e., the reduced number of steps for mobile phase preparation (water and acetonitrile R), the protection of the HPLC system by avoiding the use of phosphates for the mobile phase preparation and their precipitation during long analyses, the lower back pressure in the system, reduced consumption of reagents, and shorter turnaround time are reasons to support the use of the current optimized method for separation and quantification of the mentioned *N*-acyl-thiourea derivatives.

To demonstrate that the developed method fits its intended purpose, the validation parameters such as specificity, precision (intra-assay and intermediate precision), limits of detection (LOD) and quantification (LOQ), linearity, accuracy, and robustness have been completed, as per ICH guideline requirements [29]. The validation steps are described in the current study.

As for the specificity parameter, no interfering peaks were observed from the diluent. Although the peaks are closely chemically related, the chromatographic analytical method ensured their separation properly, avoiding a potential co-elution. The reasonable separation was sustained by resolution values higher than 2 between two consecutive peaks, along with the purity determination by purity angles < purity threshold.

The degree of repeatability was evaluated within the precision parameter. The results proved to be close to one another for six replicates of solutions (intra-assay precision), prepared on two different days (intermediate precision); specifically, the relative standard deviation values were less than 2, meeting the set acceptance criterion.

The lowest distinguished and determined concentration for every analyte in the sample was revealed by LOD—LOQ analysis, using dispersion characteristics of the regression line. For the six related compounds, the LOD limits were calculated in the range of 0.0184 µg/mL–0.0252 µg/mL, while LOQ ranged in an interval of 0.0614 µg/mL–0.0797 µg/mL. The linearity was demonstrated for each compound, covering the concentrations ranging from calculated LOQ to 6 µg/mL solutions of **1h**, **1j**, **1i**, **1m**, **1g**, and **1n**. Six regression lines corresponding to each analyte were traced for eight increasingly concentrated solutions. In every case, the analytes in the tested solutions proved to be linearly proportional to their concentration. The linearity was confirmed for every calibration plot, by evaluation of correlation coefficient, with satisfactory values (R^2^ ≥ 0.999).

The accuracy of the analytical procedure was determined over three concentration levels (80%, 100%, and 120% analytical concentrations), namely 4 µg/mL, 5 µg/mL, and 6 µg/mL solutions of **1h**, **1j**, **1i**, **1m**, **1g**, and **1n**. The standard deviation, relative standard deviation, and confidence interval were calculated for six replicate samples. The determined concentrations were reported as percent recovery of the known added amount of compound, as per ICH Q2(R1) and the results were within 98–102%.

By altering the parameters that may affect the analytical procedure, namely mobile phase composition, flow rate, and column temperature, the method proved to be robust and reliable during normal use.

The analytical method was found satisfactory for validation parameters, following ICH guidelines requirements. Hence, it can be routinely used for the simultaneous separation and quantification of compounds **1h**, **1j**, **1i**, **1m**, **1g**, and **1n** in a mixture.

## 5. Conclusions

The present study aimed to synthesize and characterize new *N*-acyl thiourea derivatives (**1a**–**1o**), incorporating both the thiazole or pyridine nucleus in the same molecule and showing a predicted antimicrobial potential previously studied by in silico approaches (the influence of electron-withdrawing, electron-donating atoms, and specific functional groups on the acyl thiourea moiety’s heterocyclic core on the biological activities have been investigated).

After synthesis, the compounds have been physiochemically characterized by their melting points, IR, NMR, and MS spectra.

The in vitro evaluation of the antimicrobial activity has been performed against problematic opportunistic Gram-positive and Gram-negative bacterial strains (*Staphylococcus aureus*, *Enterococcus faecalis*, *Escherichia coli*, *Pseudomonas aeruginosa*) in planktonic (broth microdilution assay for the determination of the MIC values) and biofilm (crystal violet microtiter assay to establish the MBIC values) growth state. Among the tested compounds, **1a**, **1g**, **1h** and **1o** displayed better activity against planktonic *Staphylococcus aureus* and *Pseudomonas aeruginosa*, as revealed by the MIC values determined by the broth microdilution assay, while **1e** revealed the lowest antibiofilm activity against *Escherichia coli*.

The thiazole ring or the pyridine nucleus substituted with halogens (**1g***: 5-chloro;*
**1h**: *2-chloro,*
**1o***: 3,5-dibromo*), attached to the molecular backbone played a moderate role in inhibiting the microorganism’s activity. The unsatisfactory inhibition of the microbial strains may be attributed to the poor penetration of the bacterial membrane. *N*-acyl thiourea derivatives may be the focus of further investigation, in which case, we aim to evaluate the molecular properties that impact the mechanism of action of the compounds against microbial strains, by designing a more advanced number of molecules so that the corroboration of the data can reach a statistically relevant estimation.

The total antioxidant activity (TAC), assessed by the DPPH method, evidenced the highest values for the compound **1i**, followed by **1a**.

Also, another purpose of our research was to optimize and validate a separation and quantification method for highly related compounds bearing a chlorine atom on the molecular backbone (**1g**, **1h**, **1i**, **1j**, **1m**, **1n**). To demonstrate that the developed method fits its intended purpose, the validation parameters such as specificity (no interfering peaks were observed from the diluent), precision (the relative standard deviation values were less than 2, meeting the set acceptance criterion), limits of detection (LOD) (calculated in the range of 0.0184–0.0252 µg/mL) and quantification (LOQ) (ranging in 0.0614–0.0797 µg/mL) interval, linearity (demonstrated for each compound, covering the concentrations ranging from calculated LOQ to 6 µg/mL solutions), accuracy (determined over three concentration levels, i.e., 80%, 100%, and 120% analytical concentrations, the results being within 98–102%), and robustness (demonstrated by altering mobile phase composition, flow rate, and column temperature parameters) have been assessed and found satisfactory, according to ICH guideline requirements. Hence, the analytical method can be routinely used for the simultaneous separation and quantification of compounds **1h**, **1j**, **1i**, **1m**, **1g**, and **1n** in a mixture.

## Data Availability

Not applicable.

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
