# Peer review of "Contribution to the Synthesis, Characterization, Separation and Quantification of New N-Acyl Thiourea Derivatives with Antimicrobial and Antioxidant Potential"

_pharmaceutics, 2023, doi:10.3390/pharmaceutics15102501_

Round 1

Reviewer 1 Report

The review manuscript entitled "Contribution to the synthesis, characterization, separation and quantification of new N-acyl thiourea derivatives with antimicrobial and antioxidant potential” from Roman et al involved the production of a new series of organic molecules, their structural characterization, biological assays (MIC, anti-biofilm activity, antioxidant activity).

Moreover, the authors developed a potential method for the separation of structural isomers.

The introduction is according to the developed topic of the manuscript, and it has bibliographical references to support the research.

Suggestion: please include current updated references between 2020 and 2023.

Considering the MIC results, it is necessary to explain why the authors decided to synthesize these molecules and how the reported previously bioinformatics design and analyses didn´t match with the lack of bioactivity. Are the MIC and antibiofilm activity results low or negligible due to the comparison to the control sample? Also, it should be noted that a deep structural analysis it is necessary to explain the design and results of these bioassays (due to the importance the authors gave to the previous design and further characterization).

Besides, the potential new methodology to separation and quantification of some similar derivatives compounds needs to be compared to others currently used (explaining the advantages or disadvantages among them).

Some points to clarify and provided a better discussion (in yellow, pdf attached):

Lines 863-865: explain how the lipophilicity changed with the position of the substituent (because it is clear that the log P changes with this structural difference)

Line 868: which is the impact for this study that the molecule didn’t obey the Lipinski’s rule considering the low bioactivity obtained?

Line 877: halogen atom (halogen radicals is incorrect because it’s corresponding to the atom with a lone electron)

Line 906: why the authors used this sentence, or which is the aim of these sentence? Compound 1i and 1n are not isomers or 1k and 1o. Please clarify.

Lines 955-956: why is the explanation that the in silico studies didn’t reflex the bioactivity results?

In addition, it is important to analyze the differences between the structures and the reported bioactivity for each pathology the authors explain because they only report data without the deep structural analyzes (according to the aim of this manuscript). I strongly suggest including a paragraph in each disease with the corresponding discussion analysis in order to understand the importance of the structural differences or similarities. Without this relevant information and analyses this manuscript only reported data as a list of molecules with the bioactivity values and one potential novel method to separate them.

Moreover, it should be necessary to remove the numbers in the molecules (schemes 1 and 3)

Furthermore, considering moving part of the tables to the support information file (table 10 to 14, please select one of them and explain the others

Additionally, check the correct nomenclature for the N atom (in italics) throughout the manuscript (in yellow, pdf attached).

Besides, for the new synthesized molecules it is necessary to show (Support information file) the NMR and IR spectra.

Considering the aim of this research it should be necessary to improve the conclusions regarding to the structural analyses the authors will develop to find an answer how each group of molecules have the specific bioactivity.

Finally, I would like to invite the authors to include the abbreviation list of words at the end of this manuscript.

Reviewer 2 Report

The proposed review article is interesting and very relevant. The authors propose a synthetic procedure for the preparation of these compounds, conduct in vitro biological studies, as well as validate methods for HPLC analysis of the substances with the highest potential activity.

Despite my positive opinion of the article, I have some critical recommendations and remarks.

The proposed review article is interesting and very relevant. The authors propose a synthetic procedure for the preparation of these compounds, conduct in vitro biological studies, as well as validate methods for HPLC analysis of the substances with the highest potential activity.

Despite my positive opinion of the article, I have some critical recommendations and remarks. Please see below:

Please check all manuscript for punctuation, grammar and spelling mistakes.

For example: Row 32 a comma is missing after 1h and before and. Please correct.

Scheme 1 Please rewrite the reaction scheme. It does look that is in pure quality (issue related to resolution). Also, please make sure that all bonds are same length. You can use ChemDraw Professional

Why the numbers of the compounds come in reverse order? Starts with 6 and ends with 1.

For all new compounds, please add 1H and 13C NMR spectra and IR spectra in the supplementary file. Then please refer them into the article.

As you work with high-resolution HRMS you need to give calculated and experimental mass, and to calculate the mass error.

Please see an example

HRMS Electrospray ionization (ESI) m/z calcd for [M+H]+ C25H21FNO3+ = 402.1500, found 402.1495 (mass error ∆m = −1.24 ppm), AND/OR calcd for [M+Na]+ C25H20FNO3Na+ = 424.1319, found 424.1313 (mass error ∆m = −1.41 ppm).

Please add the yield in grams for all compounds, after the given % yield. (in experimental part). For example:

2-((4-Ethylphenoxy)methyl)-N-((3,5-dibromopyridin-2-yl)carbamothioyl)benzamide (1o) (m.p. 140.2 – 142.9 612 0C; yield 54 %, X.XXXXg)

Overall, the article is interesting, well structured, and a lot of research has been done. In this regard, I would like to recommend after minor revision the article to be published.

The English language is fine. Minor editing is required.  

Reviewer 3 Report

A little bit more emphasis could be put on the Results description and evaluation, as there are many – not just plain tables with data to interpret by the reader. Moreover in the Discussion section the Authors discuss lipophilicity of the compounds and other aspects that were not the basis of this study. What is lacking, in my opinion, is a solid structure-activity relationship description when it comes to eg. antibacterial activity or assumed antioxidant activity (eg. what was the reference substance concentration used and what antioxidant values were obtained?). It's hard to draw clear conclusions on the new series of ligands. In fact, the numbering system of the new compounds could be simple 1-15. 

Some minor corrections that I would address:

N-acyl – shouldn’t N be italic?

line 59 - in my opinion it should be 'halo-' and 'methoxy-' 

line 62 – please use space before / after slash so as to avoid wide text gaps

line 165 -  oC – should be a degree symbol not an “0” upper case

lines 236-249 - please check text spacing

line 828 - “As the resolution values proved to be more than 2 “ – what was the basis of these findings.

Scheme 1 – please redraw the scheme, since not all of the atom letter sizes and bond angles are consistent. Moreover, the numeration of steps & substrates is misleading.

Paragraph 3.2 – please move the numerical results to Paragraph 2 – 'Materials and methods', and evaluate on the results obtained and described in paragraph 3.2.  

Scheme 2 is actually a Figure, yet this is purely semantic

Paragraph 3.3 – what are/were the standard deviations?
Paragraph 3.4. Total antioxidant capacity measurements – please evaluate on the DPPH. What does it mean that the TAC of compound is 87% - with regard to?
Table 4 – please provide units for each of shown parameters

A few typos check  and text formatting needs to be done.

Round 2

Reviewer 1 Report

The authors performed all the suggested corrections and they included the requested additional information, thanks.